# Slit-based irrigation catheters can reduce procedure-related ischemic stroke in atrial fibrillation patients undergoing radiofrequency catheter ablation

**Yun Gi Kim, Jaemin Shim**[ORCID]*****, Ki Yung Boo, Do Young Kim, Kwang-No Lee, Jong-Il Choi, Young-Hoon Kim**

Arrhythmia Center, Korea University Medicine, Seoul, Republic of Korea

* jaemins@korea.ac.kr

**Data Availability Statement:** The data set cannot be made publicly available as it contains confidential patient information. The medical law of

## Abstract

Open irrigation ablation catheters are now the standard in radiofrequency catheter ablation (RFCA) of atrial fibrillation (AF). Among various irrigation catheters, laser-cut slit-based irrigation system (Cool Flex and FlexAbility) has a unique design to cool the catheter tip more efficiently. We aimed to assess the safety of slit-based irrigation catheters regarding prevention of procedure-related ischemic complication in AF patients undergoing RFCA. The analysis was performed with Korea University Medicine Anam Hospital RFCA registry. Procedure-related ischemic complication was defined as ischemic stroke or transient ischemic attack (TIA) occurring within 30 days after RFCA. Patients were divided into 3 groups: non-irrigation, hole-based irrigation, and slit-based irrigation catheter groups. A total of 3,120 AF patients underwent first RFCA. Non-irrigation, non-slit-based irrigation, and slit-based irrigation catheters were used in 290, 1,539, and 1,291 patients, respectively. As compared with non-irrigation and non-slit-based irrigation catheter groups, slit-based irrigation catheter group had significantly older age, higher prevalence of non-paroxysmal AF, large left atrial size, and decreased left atrial appendage flow velocity. The $CHA_2DS_2$-VASc score was not different among the 3 groups. Procedure-related ischemic complication occurred in 17 patients (0.54%) with 16 ischemic strokes and 1 TIA event: 5/290 (1.72%), 11/1,539 (0.71%), and 1/1,291 (0.08%) events in non-irrigation, non-slit-based irrigation, and slit-based irrigation catheter groups, respectively (p = 0.001). Slit-based irrigation catheter was superior in direct comparison with non-slit-based irrigation catheter (0.71% vs. 0.08%; p = 0.009). Slit-based irrigation catheters were highly effective in preventing procedure-related ischemic complications.

## Introduction

Radiofrequency catheter ablation (RFCA) for symptomatic atrial fibrillation (AF) is an established treatment which is associated with significantly improved quality of life [1]. Recent clinical trials also suggest that RFCA can improve not only quality of life but also hard clinical

Republic of Korea strictly inhibits public sharing of individual patient-level data such as age, sex, body weight, or medical histories. Furthermore, Institutional Review Boards of Korea University Medical Center Anam Hospital also prohibits public sharing of individual patient-level data. However, these raw data can be provided to interested researchers upon approval of the Institutional Review Boards. The data can be requested from the Institutional Review Board of Korea University Medical Center Anam Hospital (contact via +82-02-920-6566, or +82-02-920-6086, eirbadmin@kumc.or.kr, or http://irb.kumc.or.kr).

**Funding:** This work was supported by a grant from Korea University Anam Hospital, Seoul, Republic of Korea (grant No. K1922851). https://medicine.korea.ac.kr/web/www Y. G. Kim received the fune. Funders had no role in the study design, data collection and analysis, decision to publish, or preparation of the manuscript.

**Competing interests:** The authors have declared that no competing interests exist.

**Abbreviations:** AF, atrial fibrillation; CT, computed tomography; MRI, magnetic resonance imaging; NOAC, non-vitamin K oral anticoagulant; LA, left atrium; LAA, left atrial appendage; RFCA, Radiofrequency catheter ablation; TEE, transesophageal echocardiography; TIA, transient ischemic attack.

outcomes including all cause death in patients with both AF and heart failure [2, 3]. Claim data based retrospective data also suggest that RFCA is associated with reduced risk of ischemic stroke [4, 5]. Despite various benefits of RFCA in AF patients, RFCA is associated with several catastrophic complications such as procedure-related stroke or atrio-esophageal fistula [6, 7].

Time interval between occurrence of procedure-related stroke and correct diagnosis due to patient sedation delays appropriate therapy and significant proportion of patients with procedure-related stroke have permanent neurologic consequences. Optimal treatment modalities for procedure-related stroke are also not established since majority of these strokes are due to char formation during radiofrequency energy delivery [8, 9]. Char formation during radiofrequency energy delivery is thought to be related with tissue overheating and acute cellular damage [10]. Irrigation catheters were developed in order to reduce tissue overheating and char formation [11–13]. Among irrigation catheters, open-irrigation catheters showed lower interface temperature, thrombus, and steam pop compared with closed-loop irrigation catheters indicating greater interface cooling capability [14]. Open-irrigation catheters are further classified according to their shape: 6-holes, 12-holes, 56-holes, and slit-based irrigation system. Slit-based irrigation catheters have the largest irrigation space compared to other open-irrigation catheters. However, whether slit-based irrigation catheters are superior to other open-irrigation catheters in terms of stroke prevention remains largely unknown. We aimed to compare the risk of procedure-related ischemic complication among non-irrigation, non-slit-based irrigation, and slit-based irrigation catheters.

## Methods

### Patients

Consecutive AF patients undergoing RFCA at Korea University Medicine Anam Hospital between June 1998 and April 2019 were analyzed retrospectively. A total of 3,120 patients underwent their first RFCA for AF during the study period. There was no specific exclusion criteria and all patients who underwent their first-time RFCA in our institution were included. This study was approved by the Institutional Review Board of Korea University Medicine Anam Hospital (approval number: 2020AN0165). Written informed consent was waived due to the retrospective nature of the current study. The study protocol adheres to the ethical guidelines of the 2008 Declaration of Helsinki.

### Ablation procedure and anticoagulation

The precise protocol for RFCA in our institution is published elsewhere [6]. Pre-procedural anticoagulation was performed with either warfarin or non-vitamin K oral anticoagulant (NOAC). Some patient was not prescribed with anticoagulants. However, we performed transesophageal echocardiography (TEE) in 92.9% of patients to rule out any thrombus or sludge in left atrium (LA) of left atrial appendage (LAA). Emptying, filling, and average flow velocity of the LAA were measured during TEE evaluation. Protocols of our pre-procedural imaging evaluation are published elsewhere [15]. Either computed tomography (CT) or magnetic resonance imaging (MRI) study was performed to assess the anatomy of the pulmonary veins and to create a three-dimensional reconstruction map using either EnSite NavX or CARTO systems. During RFCA, intravenous heparin was administered to maintain activated coagulation time between 300–350 seconds.

After index RFCA, anticoagulation with either warfarin or NOAC was performed for at least two months after the procedure. After two months, anticoagulation therapy was given to patients with $CHA_2DS_2$-VASc score $\geq 2$. Anticoagulation was discontinued, at the operator's

discretion, if no AF was documented on regular Holter monitoring (every 3 months for the first year and every 6 months thereafter).

## Definitions

Procedure-related ischemic complication was defined as any ischemic stroke or transient ischemic attack (TIA) which occurred within 30 days post-RFCA. Immediate procedure-related ischemic complication was defined as ischemic stroke or TIA within 3 days post-RFCA.

Ischemic stroke was defined as any neurologic symptom that persisted for more than 24 hours which could not be explained by other medical conditions. Transient ischemic attack was defined as any neurologic symptom that was not attributable to other medical cause, but resolved completely within 24 hours. If acute infarction was observed on brain CT or MRI, the event was classified as ischemic stroke despite complete restoration of neurologic symptoms. Neurologists confirmed the final diagnosis of ischemic stroke and TIA. Neurologic consequence of procedure-related stroke was classified into three stages: (i) None: no neurologic symptoms; (ii) Minimal: permanent neurologic symptoms which does not limit normal daily life; (ii) Significant: permanent neurologic symptoms which have significant limitation on normal daily life or occupation.

## Irrigation system

Irrigation system of ablation catheters were classified into three groups: non-irrigation, non-slit-based irrigation, and slit-based irrigation catheters. Non-slit-based irrigation catheters included closed-loop irrigation catheters and hole-based open-irrigation catheters. Slit-based irrigation catheters were consisted of Cool Flex and FlexAbility catheter (Abbott, Chicago, IL, USA). Flow rate of the irrigation fluid was based on the manufacturer's recommendations.

## Statistical analysis

Continuous variables are expressed as mean ± standard deviation. Categorical variables are presented as percentile value. Unpaired t-test was used to compare continuous variables. Categorical variables were compared using either the chi-square test or Fisher's exact test, as appropriate. Multivariate logistic regression analysis was performed to evaluate the impact of different type of irrigation system on procedure-related ischemic complications. Covariates were included in the multivariate model if significant difference was observed in the baseline characteristics or were a known risk factor for ischemic stroke or TIA. Statistical significance was based on p-value less than 0.05. SPSS version 24.0 (SPSS Inc., Armonk, NY, USA) was used for all statistical analyses.

## Results

### Patients

During June 1998 to April 2019, a total of 3,120 patients underwent their first RFCA for AF with 290 non-irrigation, 1,539 non-slit-based irrigation, and 1,291 slit-based irrigation ablation catheters. Baseline characteristics of the entire cohort are summarized in Table 1. Mean age was 55.74 ± 10.96 years and 78.9% were male. Previous history of thromboembolic events was observed in 8.4% of the patients and mean $CHA_2DS_2$-VASc score was 1.27 ± 1.26.

**Table 1. Baseline characteristics of the study population.**

| | Total | Procedure-related ischemic complication (-) | Procedure-related ischemic complication (+) | p value |
|---|---|---|---|---|
| | N = 3,120 | n = 3,103 | n = 17 | |
| Age (year) | 55.74 ± 10.96 | 55.72 ± 10.97 | 59.29 ± 7.94 | 0.180 |
| LA diameter (mm) | 41.20 ± 6.04 | 41.18 ± 6.04 | 44.38 ± 5.44 | 0.035 |
| Body mass index (kg/m$^2$) | 24.98 ± 3.07 | 24.98 ± 3.07 | 24.90 ± 2.41 | 0.912 |
| CHA$_2$DS$_2$-VASc | 1.27 ± 1.26 | 1.27 ± 1.27 | 1.35 ± 1.12 | 0.790 |
| LV ejection fraction (%) | 54.68 ± 6.14 | 54.68 ± 6.13 | 53.46 ± 6.84 | 0.425 |
| E/e' | 8.79 ± 3.79 | 8.78 ± 3.80 | 10.10 ± 2.28 | 0.273 |
| LAA flow velocity (cm/sec) | 48.53 ± 21.43 | 48.58 ± 21.43 | 38.80 ± 19.98 | 0.078 |
| SEC | 19.5% | 19.4% | 40.0% | 0.093 |
| Non-paroxysmal AF | 40.9% | 40.8% | 64.7% | 0.046 |
| Male sex | 78.9% | 78.9% | 82.4% | 0.725 |
| Heart failure | 6.5% | 6.5% | 5.9% | > 0.999 |
| Hypertension | 38.0% | 38.1% | 35.3% | 0.815 |
| Diabetes | 9.6% | 9.6% | 5.9% | > 0.999 |
| Stroke or TIA history | 8.4% | 8.3% | 17.6% | 0.165 |
| Vascular disease | 7.7% | 7.7% | 5.9% | > 0.999 |
| Substrate modification | 45.5% | 45.5% | 62.5% | 0.172 |
| Pre-RFCA anticoagulation | 60.4% | 60.3% | 82.4% | 0.064 |
| Post-RFCA anticoagulation | 95.0% | 95.0% | 100.0% | > 0.999 |

SEC: spontaneous echo-contrast. Other abbreviations are the same as in the text.

## Procedure-related ischemic complication

Procedure-related ischemic complication occurred in 17 patients: six patients had no permanent neurologic consequences and five patients had permanent neurologic symptoms which significantly limited their normal daily life. Mean CHA$_2$DS$_2$-VASc score was 1.35 ± 1.12 and 14 patients were male. Eleven (64.7%) patients had non-paroxysmal AF and substrate modification in addition to pulmonary vein isolation was performed in 64.7% of patients. Characteristics of these 17 patients with procedure-related stroke are summarized in Table 2. Baseline characteristics of patients with and without procedure-related stroke are compared in Table 1. Patients with procedure-related ischemic complication had larger LA diameter (44.38 ± 5.44 vs. 41.18 ± 6.04 mm; p = 0.035), higher proportion of non-paroxysmal AF (64.7% vs. 40.8%; p = 0.046).

Distribution of patients who were treated with antiplatelets, warfarin, NOAC, or no antithrombotic agents before and after RFCA is depicted in Fig 1A and 1B. Type of antithrombotic medication did not affect the incidence of procedure-related ischemic complication (p = 0.261 for pre-RFCA and 0.810 for post-RFCA; Fig 2A and 2B).

## Irrigation system

Baseline characteristics of patients who were ablated with slit-based irrigation (n = 1,291) vs. other catheters (n = 1,829) are summarized in Table 3. Those who were ablated with slit-based irrigation catheter were older (56.65 ± 10.62 vs. 55.10 ± 11.15 years; p < 0.001); had larger LA (41.63 ± 6.18 vs. 40.89 ± 5.93 mm; p = 0.001); lower LAA flow velocity (46.59 ± 21.63 vs. 50.08 ± 21.15 cm/sec; p < 0.001); and lower prevalence of spontaneous echo-contrast (17.0% vs. 21.5%; p = 0.002) and vascular disease (5.0% vs. 9.6%; p < 0.001); and higher prevalence of non-paroxysmal AF (45.5% vs. 37.7%; p < 0.001).

**Table 2. Patients who experienced peri-procedural ischemic complication.**

|  | RFCA Date | Stroke / TIA | Time to stroke or TIA (days from RFCA) | Neurologic sequela | Age | Sex | Non-Paroxysmal | CHA$_2$DS$_2$-VASc | Substrate modification | Ablation Catheter |
|---|---|---|---|---|---|---|---|---|---|---|
| 1 | 2000-04-17 | Stroke | 26 | Minimal | 61 | M | 0 | 0 | 0 | Non-irrigation |
| 2 | 2003-03-14 | Stroke | 28 | Significant | 53 | F | 0 | 2 | 0 | Non-irrigation |
| 3 | 2005-11-14 | Stroke | 3 | Minimal | 42 | M | 1 | 0 | 1 | Non-irrigation |
| 4 | 2005-12-08 | Stroke | 1 | Minimal | 66 | M | 1 | 2 | 1 | Non-irrigation |
| 5 | 2006-02-20 | Stroke | 4 | None | 52 | M | 1 | 3 | 1 | Non-irrigation |
| 6 | 2006-03-15 | Stroke | 1 | Minimal | 70 | M | 0 | 2 | 0 | Chilli |
| 7 | 2006-05-29 | Stroke | 1 | Significant | 60 | M | 1 | 0 | 1 | Chilli |
| 8 | 2008-12-10 | Stroke | 9 | None | 72 | F | 0 | 2 | 0 | Celsius |
| 9 | 2011-02-21 | Stroke | 0 | Significant | 67 | M | 0 | 1 | 0 | Navistar |
| 10 | 2012-06-07 | Stroke | 3 | None | 50 | M | 1 | 2 | 1 | Celsius |
| 11 | 2013-05-28 | Stroke | 0 | Significant | 55 | M | 1 | 0 | 1 | Thermocool |
| 12 | 2013-06-18 | Stroke | 0 | Minimal | 57 | M | 1 | 0 | 1 | Thermocool |
| 13 | 2017-04-06 | Stroke | 0 | Minimal | 57 | M | 1 | 1 | 1 | SmartTouch |
| 14 | 2017-04-13 | Stroke | 1 | None | 60 | M | 1 | 1 | 1 | SmartTouch |
| 15 | 2017-10-18 | TIA | 0 | None | 70 | F | 0 | 3 | 0 | TactiCath |
| 16 | 2018-08-16 | Stroke | 1 | None | 58 | M | 1 | 3 | 1 | Cool Flex |
| 17 | 2019-02-14 | Stroke | 0 | Significant | 58 | M | 1 | 1 | 1 | TactiCath |
|  |  |  |  |  | 59.29 ± 7.94 | 82.4% | 64.7% | 1.35 ± 1.12 | 64.7% |  |

Abbreviations are the same as in the text.

One procedure-related ischemic complication occurred in patients who were ablated with slit-based irrigation (0.08%) whereas 16 events occurred in patients ablated with other catheters (0.87%; p = 0.003; Fig 3A). When classified into three groups, non-irrigation catheters had highest incidence of procedure-related ischemic complication (1.72%) followed by non-slit-based irrigation catheters (0.71%) and slit-based irrigation catheters (0.08%) (p = 0.001; Fig 3B). Slit-based irrigation catheter was also superior in direct comparison with non-slit-based irrigation catheter (0.71% vs. 0.08%; p = 0.009). Incidence of immediate procedure-related ischemic complication (occurring within 3 days of post-RFCA) also differed significantly according to catheter type (Fig 3C and 3D). The results were identical when closed-loop irrigation catheters were classified as non-irrigation catheters (S1 Fig). Multivariate model revealed that use of slit-based irrigation catheter was the only significant predictor of procedure-related ischemic complication (Table 4). The influence of chronological variations in the use of different type of ablation

**A    Pre-RFCA anticoagulation**

**B    Post-RFCA anticoagulation**

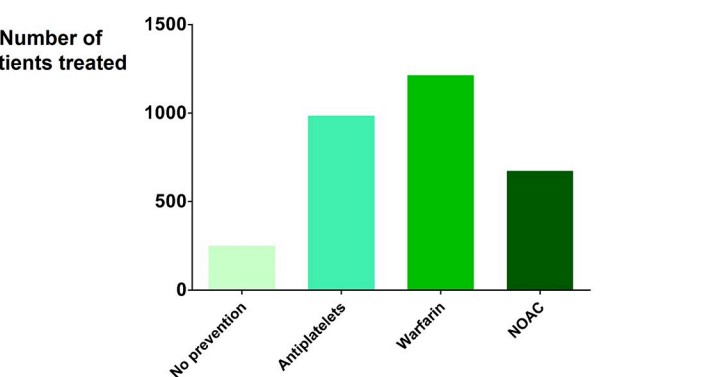

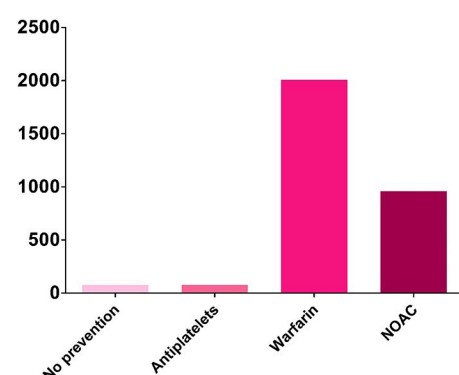

**Fig 1. Pre- and post-RFCA antithrombotic therapy.** (A) Type of antithrombotic therapy received before RFCA. (B) Type of antithrombotic therapy received after RFCA. Abbreviations are the same as in the text.

catheters were also adjusted and type of ablation catheter remained the only independent variable for procedure-related ischemic complication (S1 and S2 Tables).

## Discussion

This study revealed that slit-based irrigation catheters can be effective in preventing procedure-related ischemic complication. Non-irrigation catheters showed highest risk for procedure-related ischemic complication and peri-procedural anticoagulation did not influence the risk for procedure-related ischemic complication. This data is the largest study to date analyzing safety issue of different irrigation systems and the first report demonstrating that irrigation systems can affect actual clinical stroke and TIA.

### Ischemic complication during RFCA

Together with atrio-esophageal fistula, peri-procedural ischemic complication is most dreaded complication in AF patients undergoing RFCA [16, 17]. Since patients are in deep sedation or

**A    Pre-RFCA anticoagulation**

**B    Post-RFCA anticoagulation**

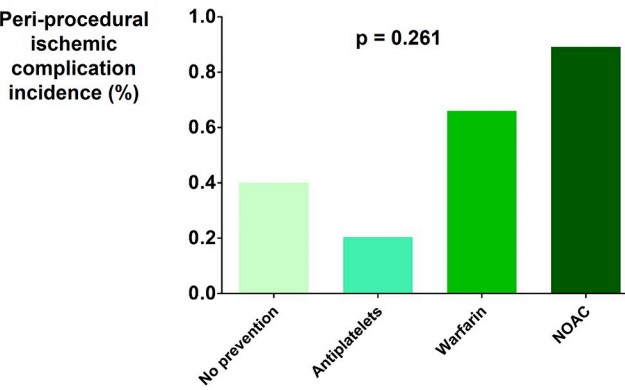

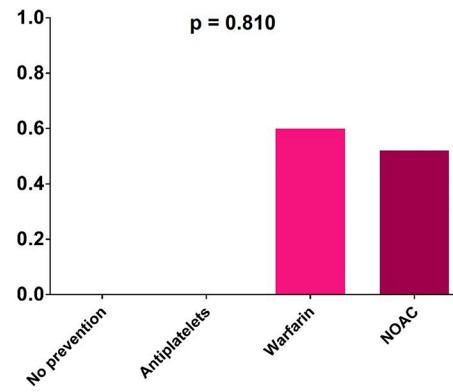

**Fig 2. Incidence of peri-procedural ischemic complication according to antithrombotic therapy.** (A) Incidence of peri-procedural ischemic complication according to type of antithrombotic therapy received before RFCA. (B) Incidence of peri-procedural ischemic complication according to type of antithrombotic therapy received before RFCA. Abbreviations are the same as in the text.

Table 3. Baseline characteristics of the study population according to ablation catheter used.

| | Slit-based Irrigation | Other catheters | p value |
|---|---|---|---|
| | n = 1,291 | n = 1,829 | |
| Age (years) | 56.65 ± 10.62 | 55.10 ± 11.15 | < 0.001 |
| LA diameter (mm) | 41.63 ± 6.18 | 40.89 ± 5.93 | 0.001 |
| Body mass index (kg/m$^2$) | 25.01 ± 3.10 | 24.96 ± 3.05 | 0.608 |
| CHA$_2$DS$_2$-VASc | 1.25 ± 1.22 | 1.29 ± 1.29 | 0.425 |
| LV ejection fraction (%) | 54.52 ± 5.93 | 54.80 ± 6.28 | 0.213 |
| E/e' | 8.78 ± 3.27 | 8.79 ± 4.18 | 0.931 |
| LAA flow velocity (cm/sec) | 46.59 ± 21.63 | 50.08 ± 21.15 | < 0.001 |
| SEC | 17.0% | 21.5% | 0.002 |
| Non-paroxysmal AF | 45.5% | 37.7% | < 0.001 |
| Male sex | 78.9% | 78.9% | > 0.999 |
| Heart failure | 5.9% | 7.0% | 0.216 |
| Hypertension | 36.9% | 38.8% | 0.289 |
| Diabetes | 9.0% | 10.1% | 0.316 |
| Stroke or TIA history | 8.7% | 8.1% | 0.599 |
| Vascular disease | 5.0% | 9.6% | < 0.001 |
| Substrate modification | 44.9% | 46.0% | 0.562 |
| Pre-RFCA anticoagulation | 67.3% | 55.6% | < 0.001 |
| Post-RFCA anticoagulation | 97.2% | 93.5% | < 0.001 |

SEC: spontaneous echo-contrast. Other abbreviations are the same as in the text.

under general anesthesia, they cannot complain any neurologic symptoms when their cerebral arteries are occluded by thrombus or char. It takes several hours to complete the procedure and to recover from sedation or general anesthesia. Therefore, immediate revascularization was often not feasible. Furthermore, char formation during radiofrequency energy delivery makes harder thrombus material which makes revascularization therapy even more difficult. Although massive thrombus and char embolization during RFCA is a rare event, it is usually unrecoverable once occurred.

In our registry, five patient had significant neurologic sequelae defined as permanent neurologic symptoms limiting their normal daily life or occupation. Making ischemic stroke during treatment process of AF is an irony and 0.16% (5 among 3,120 patients) risk of having significant neurologic sequelae cannot be ignored.

## Irrigation system

Local temperature elevation at the distal tip of ablation catheter during RFCA can result in temperature-dependent coagulum and char formation [10]. Histological review of these deposits revealed that it was consisted of denaturized and aggregated proteins rather than classical thrombus [10]. Therefore, reducing the temperature of distal electrode tip can potentially reduce char and thrombus formation.

In our registry, patients who were ablated with non-irrigation catheters showed highest risk of having procedure-related ischemic complication (1.72%) followed by non-slit-based irrigation (0.71%) and slit-based irrigation catheters (0.08%). Irrigation catheters were developed for two major reasons: formation of deeper lesions and reduction of ischemic complications [11]. Previous studies demonstrated that irrigation catheters were capable of making deeper lesions as compared with non-irrigation catheters [18, 19]. Although irrigation catheters have

**A**  **Procedure-related ischemic complication**
**(≤ 30 days)**

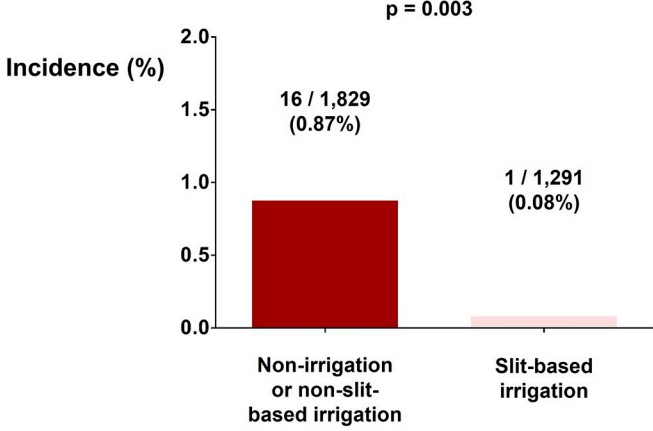

**B**  **Procedure-related ischemic complication**
**(≤ 30 days)**

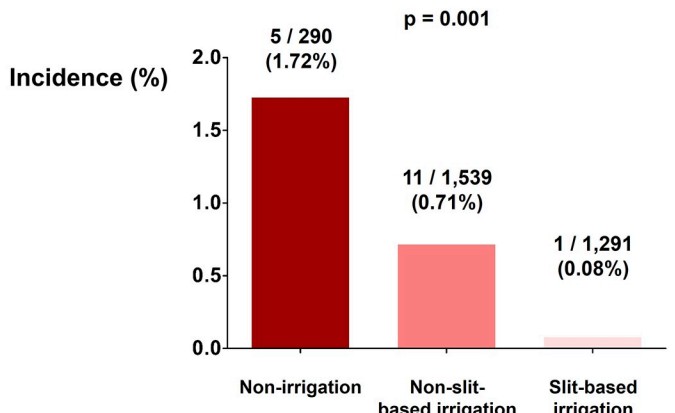

**C**  **Immediate procedure-related ischemic complication**
**(≤ 3 days)**

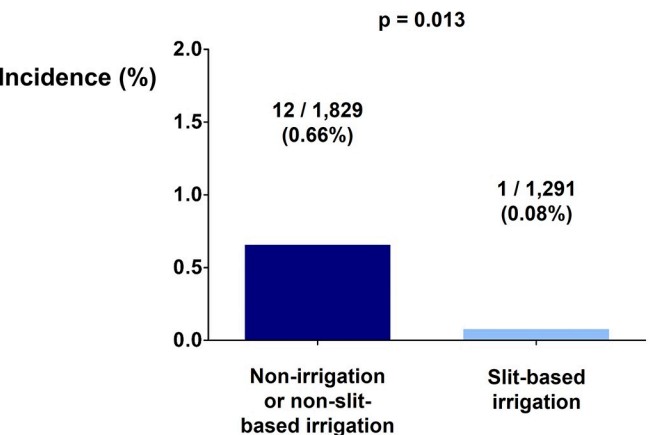

**D**  **Immediate procedure-related ischemic complication**
**(≤ 3 days)**

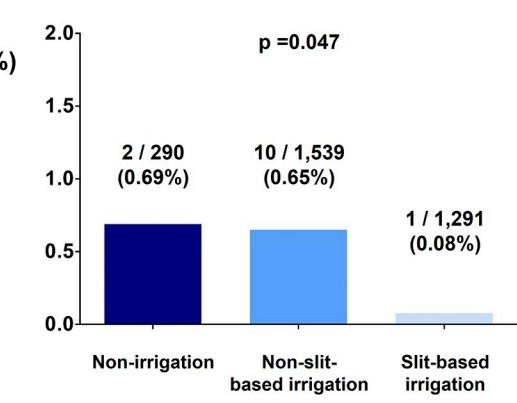

**Fig 3. Incidence of peri-procedural ischemic complication according to catheter type.** (A, B) Incidence of peri-procedural ischemic complication according to catheter type. (C, D) Incidence of immediate peri-procedural ischemic complication according to catheter type.

shown that coagulum formation can be reduced as compared to non-irrigation catheters, whether irrigation catheters can reduce clinical ischemic complications remains largely unknown [11].

Previous study reported that irrigated tip catheter did not reduce the incidence of symptomatic peri-procedural ischemic stroke [20]. However, the study enrolled patients from February 2001 to January 2008 where no slit-based irrigation catheters were available and our study demonstrated a lower incidence of procedure-related ischemic complication in slit-based irrigation catheters. Our study also showed lower incidence of procedure-related ischemic complication in non-slit-based irrigation catheters compared non-irrigation catheters (0.71% vs. 1.72%). The discrepancy between the two studies can be due to small sample size or different type of non-irrigation catheter (predominantly 8mm tip in the study by Scherr et al. and 4mm tip in ours).

**Table 4. Multivariate model for procedure-related ischemic complication.**

|  | Model 1 (event n = 15) | | Model 2 (event n = 17) | |
|---|---|---|---|---|
|  | OR (95% CI) | p value | OR (95% CI) | p value |
| Slit-based irrigation catheter | 0.073 (0.009–0.562) | 0.012 | 0.070 (0.009–0.536) | 0.010 |
| Age (year) | 1.019 (0.962–1.079) | 0.525 | 1.041 (0.989–1.097) | 0.123 |
| Sex | 1.158 (0.306–4.380) | 0.829 | 1.398 (0.389–5.032) | 0.608 |
| Heart failure | 0.535 (0.064–4.482) | 0.564 | 0.642 (0.083–4.977) | 0.672 |
| Hypertension | 0.653 (0.208–2.056) | 0.467 | 0.737 (0.258–2.108) | 0.570 |
| Diabetes mellitus | 0.634 (0.081–4.980) | 0.665 | 0.532 (0.069–4.109) | 0.545 |
| Previous ischemic stroke/TIA | 1.732 (0.467–6.418) | 0.411 | 1.504 (0.412–5.484) | 0.536 |
| Vascular disease | 0.562 (0.071–4.466) | 0.586 | 0.472 (0.060–3.685) | 0.474 |
| Non-paroxysmal AF | 2.011 (0.462–8.747) | 0.352 | 2.388 (0.678–8.409) | 0.175 |
| Substrate modification | 0.732 (0.190–2.829) | 0.651 | 0.796 (0.245–2.587) | 0.704 |
| Left atrial diameter (mm) | 1.042 (0.945–1.149) | 0.410 | Not included | Not included |
| Flow velocity of Left atrial appendage (cm/sec) | 0.992 (0.960–1.025) | 0.624 | Not included | Not included |
| SEC | 1.385 (0.425–4.514) | 0.589 | Not included | Not included |
| Pre-RFCA anticoagulation | 5.151 (0.623–42.606) | 0.128 | 2.375 (0.612–9.213) | 0.211 |

OR: odds ratio. Other abbreviations are the same as in the text.

## Anticoagulation

Previous studies reported that continuation of anticoagulation in peri-operative period can reduce ischemic complication after RFCA [21, 22]. However, pre- or post-RFCA anticoagulation did not affect the incidence of peri-procedural ischemic complication in our study. Since embolic material formed by catheter ablation is mainly consisted of denaturized and aggregated proteins rather than classical thrombus, peri-procedural ischemic complication can still occur despite optimal anticoagulation therapy [10]. Indeed, heat induced protein denaturation and aggregation occurred independently from heparin concentration [10]. Nevertheless, optimal anticoagulation therapy should be done in peri-procedural period since denaturized material formed by radiofrequency energy can activate coagulation cascade. Both irrigation and anticoagulation are essential.

## Limitations

Our results are based on retrospective analysis and therefore, are not free from intrinsic limitation of such analysis. Differences in baseline characteristics such as prevalence of vascular disease, AF type, LA diameter, and LAA flow velocity were observed between slit-based vs. non-slit based catheter groups although it was adjusted in the multivariate model. Unmeasured confounders might exist in our cohort. Systematic variations in the ascertainment of stroke or TIA over time might exist since our cohort extends more than 20 years. However, all stroke or TIA events were diagnosed by neurologists and this was centrally reviewed during data collection process. Although total patient number was large, the number of peri-procedural ischemic complication was small limiting statistical power. Slit-based irrigation catheters do not have contact-force sensing capability. Therefore, ablation parameters such as ablation index or lesion size index which has shown to improve ablation outcome cannot be utilized when using slit-based irrigation catheters [23, 24]. This can be a critical limitation of slit-based irrigation catheters and contact-force sensing ability should be added in the near future.

## Conclusion

Slit-based irrigation catheters may reduce the risk of peri-procedural ischemic complication compared with non-irrigation or non-slit-based irrigation catheters. Our results should be tested in future clinical trials.

## Supporting information

**S1 Fig. Incidence of peri-procedural ischemic complication according to catheter type.** (PDF)

**S1 Table. Chronological difference among use of different catheter types.** (PDF)

**S2 Table. Multivariate model for procedure-related ischemic complication.** (PDF)

## Author Contributions

**Conceptualization:** Yun Gi Kim, Jaemin Shim, Young-Hoon Kim.

**Data curation:** Yun Gi Kim, Do Young Kim, Kwang-No Lee.

**Formal analysis:** Yun Gi Kim, Jaemin Shim, Ki Yung Boo, Do Young Kim, Kwang-No Lee.

**Funding acquisition:** Yun Gi Kim.

**Investigation:** Yun Gi Kim, Jaemin Shim.

**Methodology:** Yun Gi Kim, Jaemin Shim, Ki Yung Boo, Do Young Kim, Kwang-No Lee.

**Project administration:** Yun Gi Kim, Kwang-No Lee.

**Resources:** Yun Gi Kim.

**Software:** Yun Gi Kim.

**Supervision:** Jaemin Shim, Jong-Il Choi, Young-Hoon Kim.

**Validation:** Yun Gi Kim, Jaemin Shim, Ki Yung Boo, Jong-Il Choi, Young-Hoon Kim.

**Visualization:** Yun Gi Kim.

**Writing – original draft:** Yun Gi Kim, Jaemin Shim, Young-Hoon Kim.

**Writing – review & editing:** Yun Gi Kim, Jaemin Shim, Jong-Il Choi, Young-Hoon Kim.

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
