## [Decision Letter · Decision Letter 0]

20 Jul 2020

PONE-D-20-15554

Slit-based Irrigation Catheters can Reduce Procedure-related Ischemic Stroke in Atrial Fibrillation Patients Undergoing Radiofrequency Catheter Ablation

PLOS ONE

Dear Dr. Shim

Thank you for submitting your manuscript to PLOS ONE. After careful consideration, we feel that it has merit but does not fully meet PLOS ONE’s publication criteria as it currently stands. Therefore, we invite you to submit a revised version of the manuscript that addresses the points raised during the review process.

We look forward to receiving your revised manuscript.

Kind regards,

Giuseppe Coppola

Academic Editor

PLOS ONE

Journal Requirements:

2. In your Methods section, please state the date(s) upon which the data source was accessed.

Reviewers' comments:

Reviewer's Responses to Questions

**Comments to the Author**

1. Is the manuscript technically sound, and do the data support the conclusions?

Reviewer #1: Partly

Reviewer #2: Partly

Reviewer #3: No

2. Has the statistical analysis been performed appropriately and rigorously? 

Reviewer #1: Yes

Reviewer #2: I Don't Know

Reviewer #3: Yes

3. Have the authors made all data underlying the findings in their manuscript fully available?

Reviewer #1: Yes

Reviewer #2: Yes

Reviewer #3: Yes

4. Is the manuscript presented in an intelligible fashion and written in standard English?

Reviewer #1: Yes

Reviewer #2: Yes

Reviewer #3: Yes

5. Review Comments to the Author

Reviewer #1: The manuscript is quite interesting, BUT there is a big problem between what it is stated in the text and what one can read in table 3. For instance, in the text it is said that "slit - based" had "higher prevalence of SEC", but in table 3 incidence in slit based is 17% and in other is 21.5% (exatcly the opposite!).

Another important issue is that "other catheters" were less anticoagulated BEFORE ablation (55.6% vs 67.3%) and remained less anticoagulated AFTER ablation (93.5% vs 97.2%) and both values were statistically signiticant. Finally, incidence of vascular diseases was almost twice as high in "other catheters" group. This point should be highlighted in the discussion

Thus, or there is some mistake in the table or this could represent an important bias in judging superiority of the slit-based catheters. Authors must explain this discrepance.

Reviewer #2: Kim et al present the results of a retrospective study of peri-procedural ischemic stroke in 3120 patients undergoing first RF catheter ablation at a single center over 20 years. They conclude that use of slit-based irrigated catheters was associated with a lower incidence of ischemic stroke, and this was independent of other risk factors in multivariate analysis.

Questions and comments:

--How do the authors account for the possibility of changes in operative technique over the 20+ year period of analysis? If non-irrigated catheters were used predominantly in the early years, followed by non-slit-based irrigated catheters, followed by irrigated catheters, there might have been other differences that were not accounted for but correlated with the type of irrigation. For example, was the ACT target identical from 1998 to 2009? Was the timing of the heparin bolus the same, or did operators wait until after transseptal access was achieved in the early years? Was perioperative oral anticoagulation completely constant over this long time period? Were any changes made to sheath aspiration/flush technique, or number/type of diagnostic catheters? How many ablation catheters incorporated contact-force sensing? Was there a difference in warfarin vs NOAC use?

--Why do authors lump together internally irrigated catheters and external non-slit-based irrigation catheters? These are very different designs, in fact more dissimilar than slit-based and non-slit-based irrigation catheters, and it seems unwarranted to include them in a single group for analysis.

--How many of the 17 ischemic complications occurred >24 hours after the ablation procedure?

--What proportion of patients in the non-irrigated, non-slit-based irrigated and slit-based irrigated groups had substrate ablation in addition to PVI? Why was substrate ablation not included as a variable in the multivariate model presented in Table 4?

--The text says the slit-based irrigated group had higher prevalence of SEC, but Table 3 says it was lower. Which is correct? Why was SEC not included as a variable in the multivariate model presented in Table 4?

--Since irrigated catheters are no longer used for AF ablation, and the slit-based irrigated catheters analyzed here (CoolFlex and Flexability) do not incorporate contact-force sensing, the Discussion should include some comment on the trade-off of giving up contact-force sensing for slit-based irrigation, including new lesion metrics (Ablation Index, Lesion Size Index) that rely on these catheters.

Reviewer #3: The authors use a retrospective analysis of a single center large database to provide data that slit-like irrigation is associated with reduced rates of thromboembolic complications following AF ablation compared to other irrigated and non-irrigated catheters. The overall number of patients undergoing ablation is large, although this database spans 20 years of practice with a number of systematic changes over time. Stroke or TIA occurred significantly less with irrigated catheters than non-irrigated catheters. More importantly, slit-like irrigation appeared to be associated with less events compared to non-slit irrigation catheters despite similar number of cases. If true, this is an important observation.

The major limitation is the contemporaneous changes in clinical practice, especially anticoagulation during the 20 years of analysis. As practice evolves, often a whole-system change occurs at a center making one catheter and non-fluoroscopic mapping system dominant for a particular time. This can make an untended impact on outcomes in a retrospective analysis like this. Specifically, the introduction and use of uninterrupted anticoagulation during this period. Specifically, the number of patients undergoing ablation with uninterrupted anticoagulation needs to be very similar between groups of irrigated catheters for any confidence that there is a real difference in the irrigation architecture as a mechanism of the difference in outcome. If this cannot be assured, it becomes very difficult to know if the difference should drive changes in practice.

The number of outcomes are very small. As such, a small difference in confounding practice (like OAC use during the procedure) can have a major impact.

As such, the authors need to make it clear if the catheter types, especially the different irrigated catheters, were used at the same time or overlapping time periods. More importantly, a clear analysis of the number of patients undergoing uninterrupted OAC is required.

It is recognized that the slit irrigation group had worse risk factors that would confound by making the thromboembolic rates higher in this group. It might also make the authors more likely to use uninterrupted OAC for this group.

There are other limitations in this kind of analysis that cannot be clarified including possible systematic differences in the ascertainment of stroke over time in a database.

What specific definition of stroke was used as opposed to TIA and mimickers like migraine? Ordinarily TIA is not used as a hard endpoint, although there is only one TIA among the outcomes.

Coolflex catheters do not have force measuring capabilities. As such, authors have traded off this function for different irrigation. This raises the importance of the veracity of this data as a change to a slit irrigation ablation eliminates an ablation technology important at many centers.

6. PLOS authors have the option to publish the peer review history of their article (what does this mean?). If published, this will include your full peer review and any attached files.

Reviewer #1: **Yes: **Giosue Mascioli, MD; FAIAC; FEHRA, FESC

Reviewer #2: No

Reviewer #3: No

---

## [Author Response · Author response to Decision Letter 0]

19 Aug 2020

We appreciate your time and effort to review our manuscript. 

We have uploaded a separate file containing our response to your comments. 

Thank you. 

Sincerely, 

Jaemin Shim. MD.

---

## [Decision Letter · Decision Letter 1]

4 Sep 2020

Slit-based Irrigation Catheters can Reduce Procedure-related Ischemic Stroke in Atrial Fibrillation Patients Undergoing Radiofrequency Catheter Ablation

PONE-D-20-15554R1

Dear Dr.  Shim,

We’re pleased to inform you that your manuscript has been judged scientifically suitable for publication and will be formally accepted for publication once it meets all outstanding technical requirements.

Kind regards,

Giuseppe Coppola

Academic Editor

PLOS ONE

Additional Editor Comments (optional):

Reviewers' comments:

Reviewer's Responses to Questions

**Comments to the Author**

1. If the authors have adequately addressed your comments raised in a previous round of review and you feel that this manuscript is now acceptable for publication, you may indicate that here to bypass the “Comments to the Author” section, enter your conflict of interest statement in the “Confidential to Editor” section, and submit your "Accept" recommendation.

Reviewer #1: All comments have been addressed

Reviewer #2: All comments have been addressed

2. Is the manuscript technically sound, and do the data support the conclusions?

Reviewer #1: Yes

Reviewer #2: Partly

3. Has the statistical analysis been performed appropriately and rigorously? 

Reviewer #1: Yes

Reviewer #2: Yes

4. Have the authors made all data underlying the findings in their manuscript fully available?

Reviewer #1: Yes

Reviewer #2: Yes

5. Is the manuscript presented in an intelligible fashion and written in standard English?

Reviewer #1: Yes

Reviewer #2: Yes

6. Review Comments to the Author

Reviewer #1: The Authors adequately adressed all the issues elicited by the reviewers. Being a retrospective study, not all bias can be solved, but anyhow the results presented after this new analysis are interesting and convincing. In my opinion the manuscript can now be published as it is.

Reviewer #2: The authors have submitted a substantially improved version of this manuscript that adequately addresses my concerns.

7. PLOS authors have the option to publish the peer review history of their article (what does this mean?). If published, this will include your full peer review and any attached files.

Reviewer #1: **Yes: **Giosue Mascioli

Reviewer #2: No

---

## [Editor Report · Acceptance letter]

22 Sep 2020

PONE-D-20-15554R1 

Slit-based Irrigation Catheters can Reduce Procedure-related Ischemic Stroke in Atrial Fibrillation Patients Undergoing Radiofrequency Catheter Ablation 

Dear Dr. Shim:

I'm pleased to inform you that your manuscript has been deemed suitable for publication in PLOS ONE. Congratulations! Your manuscript is now with our production department. 

Kind regards, 

on behalf of

Dr. Giuseppe Coppola 

Academic Editor

PLOS ONE